# Improved Random Forest for the Automatic Identification of *Spodoptera frugiperda* Larval Instar Stages

**Jiajun Xu [1], Zelin Feng [1], Jian Tang [2], Shuhua Liu [2], Zhiping Ding [3], Jun Lyu [1], Qing Yao [1] and Baojun Yang [2,*]**

[1] School of Computer Science and Technology, Zhejiang Sci-Tech University, Hangzhou 310018, China
[2] State Key Laboratory of Rice Biology, China National Rice Research Institute, Hangzhou 311499, China
[3] Zhangjiagang Customs, Zhangjiagang 215623, China
[*] Correspondence: yangbaojun@caas.cn

**Abstract:** *Spodoptera frugiperda* (fall armyworm, FAW) is a global agriculture pest. Adults have a strong migratory ability and larvae feed on the host stalks, which pose a serious threat for maize and other crops. Identification and counting of different instar larvae in the fields is important for effective pest management and forecasting emergence and migration time of adults. Usually, the technicians identify the larval instars according to the larva morphological features with the naked eye or stereoscope in the lab. The manual identification method is complex, professional and inefficient. In order to intelligently, quickly and accurately identify the larval instar, we design a portable image acquisition device using a mobile phone with a macro lens and collect 1st-6th instar larval images. The YOLOv4 detection method and improved MRES-UNet++ segmentation methods are used to locate the larvae and segment the background. The larval length and head capsule width are automatically measured by some graphics algorithms, and the larval image features are extracted by SIFT descriptors. The random forest model improved by Boruta feature selection and grid search method is used to identify the larval instars of FAWs. The test results show that high-definition images can be easily collected by using the portable device (Shenzhen, China). The MRES-UNet++ segmentation method can accurately segment the larvae from the background. The average measurement error of the head capsule width and body length of moth larvae is less than 5%, and the overall identification accuracy of 1st–6th instar larvae reached 92.22%. Our method provides a convenient, intelligent and accurate tool for technicians to identify the larval instars of FAWs.

**Keywords:** *Spodoptera frugiperda*; larval instar; automatic identification; background segmentation; improved random forest; Boruta feature selection

## 1. Introduction

*Spodoptera frugiperda* (fall armyworm, FAW) has become a major global agricultural pest according to the Food and Agriculture Organization of the United Nations (FAO) [1–3]. The FAW shows strong migratory, dispersal, and productive abilities. The FAW can migrate over long distances with the wind and the larvae have the characteristics of wide suitable areas, overeating damage and strong insecticide resistance [4,5]. The larvae feed on the leaves, stems and reproductive organs of 186 plant species, and cause economic losses to maize, rice, sorghum, sugarcane, wheat and vegetable crops [6]. Real-time and accurate monitoring of FAWs in fields is necessary to ensure accurate forecasting and early warning, and to reduce its yield loss [7]. Usually, population monitoring of lepidoptera insect pests in the fields involves all development stages of insect pests, including the egg, different instars of larvae, pupa and adults. Among them, identification and counting of different instar larvae in the fields is very important for the effective pest management and forecasting emergence and migration time of adults. In the larval stage of the *S. frugiperda*, the larval motion range is limited, and the distribution in the field is relatively centralized, which can reduce the cost of prevention in this period. Through the analysis of the larval instars,

it is helpful to find out their growth law and release the early warning information of pests in advance so as to carry out the corresponding measures of prevention and control. However, the identification of larval instars needs the higher professional technicians to identify them according to the morphological features and size of the pest larvae, which is time-consuming and complicated [8].

The FAW larvae have a typical growth and development pattern of lepidopteran larvae, that is, the periodic molting. Usually, the identification method of lepidopteran larval instars relies on the technicians to visually evaluate them in fields or use the stereoscope to measure the head capsule width as the main or even the sole judgement criterion in labs. The length, the colors and the patches of the larval head, chest and abdomen of larvae are often taken as the supplementary reference information [9]. Some insect larvae show some significant characteristic changes in different instars, which make it easy to distinguish the larval instars and to obtain a higher accuracy rate [10,11]. The FAW larvae may feed on different crops under different field conditions. It results in the different growth rates and individual variation, which makes it difficult to accurately determine the larval instars only by a single feature, such as head capsule width, body length and body color [12].

With the rapid development of image processing technology and artificial intelligence, some progress has been made on image-based pest and larval identification [13–18]. Ye et al. [19] used the ResNet-Locust-BN model to identify two locust species and instars. A Batch Normalization (BN) layer was added before the convolutional layer for feature normalization, and the learning rate and activation function were adjusted. The overall accuracy of this model in identifying East Asian migratory locust (3rd instar, 5th instar, adult) was 90.3%. Johari et al. [20] used a hyperspectral camera to collect images of the second to fifth instar larvae of the bagworm indoors, and used the threshold segmentation method to separate the worms. Then, morphological features such as wavelength, spectral reflectance, insect length, and insect body area in different spectral regions were extracted to identify larval instars. Zhang et al. [21] performed the 3D imaging by scanning wheat grains infected by rice weevil, and used the optimized support vector machine algorithm to identify the instar of rice weevils. The machine-based algorithm could identify the instars of grain pests, and the classification accuracy rate of young and old larvae was 95%, which has a certain improvement in method compared with manual morphological index analysis. However, the above methods also have some limitations. Larval instar identification methods based on deep learning often require a lot of calculation costs, and does not have good explanatory power; the larval instar identification method based on 3D imaging requires the expensive hardware equipment and its portability is poor, which makes it difficult to operate in the field.

To solve the problem mentioned above, we propose the corresponding solutions. To easily collect the images of pests, we design a portable larval image capturing device to collect the larval images of the *S. frugiperda*. Meanwhile, to automatically identify the larval instars of FAWs, we propose an automatic larval instar identification method based on improved random forest, which mainly uses the parameter optimization strategy of RandomizedSearchCV [22] and GridSerachCV [23] to obtain the best n_ estimators and max_ depth for getting the model of optimal auc and average identification accuracy. In addition, in order to realize the automatic calculation of larval body length, head capsule width and other morphological characteristics, we propose an improved larval automatic segmentation algorithm – MRES-UNet++. The self-attention module, atrous convolution and bicubic interpolation methods are used to optimize the UNet++ network to improve the MIoU (Mean Intersection over Union) and edge smoothness.

## 2. Materials and Methods

### 2.1. Identification Pipeline of Larval Instars

The automatic identification pipeline of larval instars of FAW is shown in Figure 1. The self-designed portable image capturing device is used to collect larva images of FAWs. The 1st–6th instar larvae in images are located and segmented. The body length and head

capsule width of the larvae are automatically measured and the local features of the larvae are extracted. The larval instar of *S. frugiperda* is automatically identified by the improved random forest model.

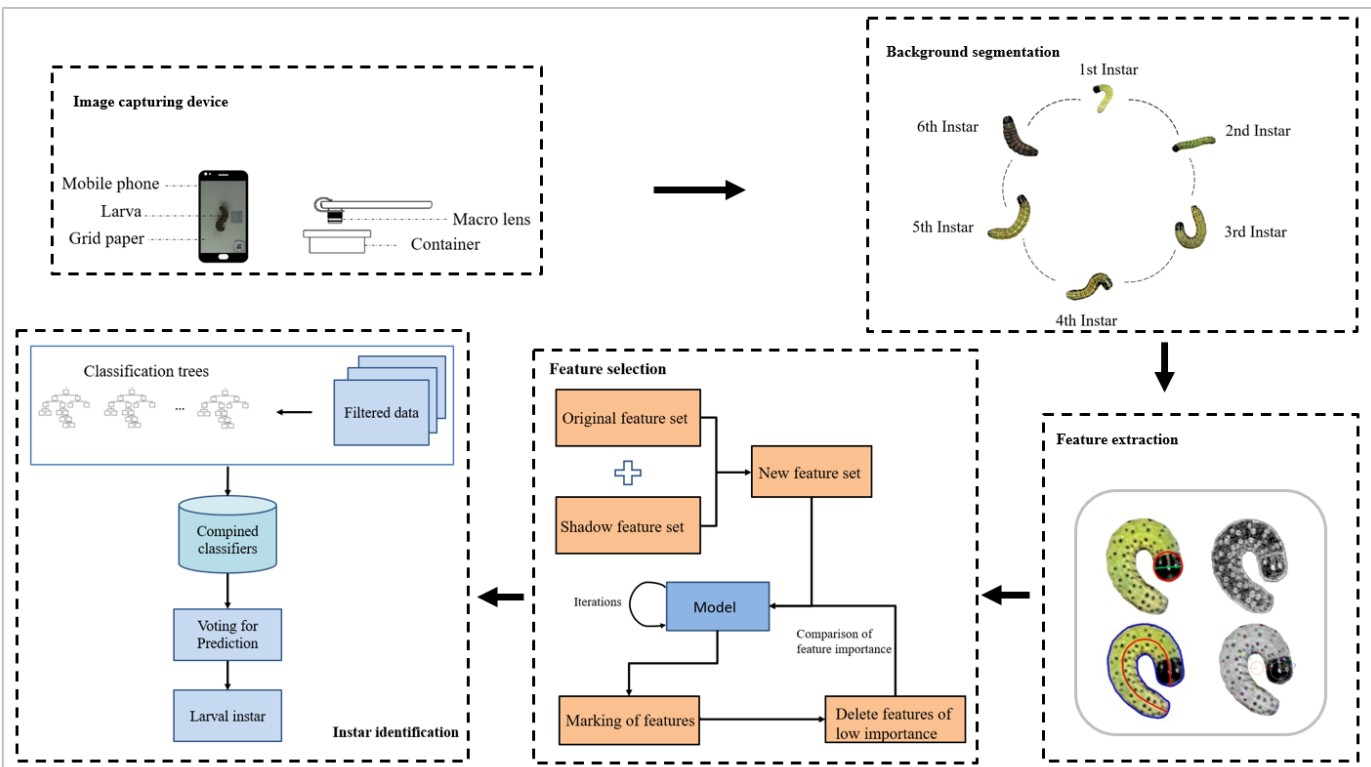

**Figure 1.** The identification pipeline of FAW larval instars.

## 2.2. Collection of Larva Images

The image capturing device used in this paper is mainly composed of a container for placing the larvae and grid paper, a macro lens (APEXEL, Shenzhen in China), and a mobile phone. The grid paper is marked with a 10 mm × 10 mm grey square block as the criteria for calculating the larval geometric size. The macro lens adopts 4.3× magnification, which is installed on the camera of the phone to magnify the larva in the image.

A total of 210 newly hatched larvae of *S. frugiperda* were separately placed in a feeding tray filled with fresh corn leaves, and kept in a growth chamber under 25 ± 1 °C and 70 ± 10% relative humidity. All the larvae after each molting were put on the grid paper, which was placed at the bottom of the square container for collecting larva images. The larvae were then placed back on the original feeding tray for feeding. Fresh leaves were added regularly until the larvae pupate. The molting date was recorded as the next instar. There were six instars in total under this experimental condition (Figure 2).

With this image capturing device, 1376 FAW larval images were taken in total, and the training set and test set were divided according to the ratio of 7:3 for training and testing images.

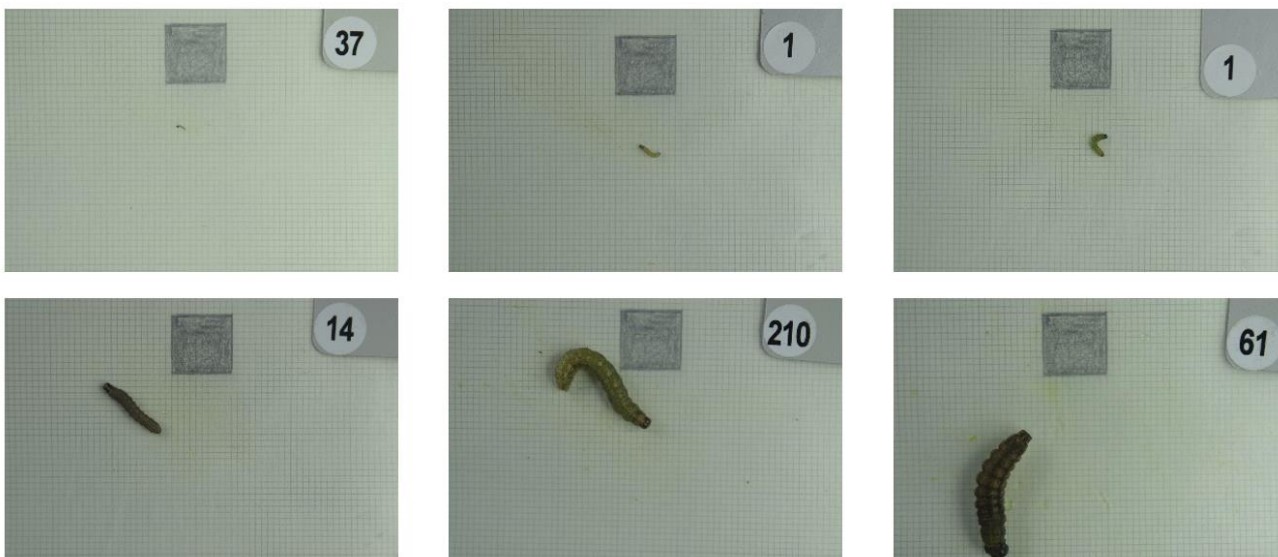

**Figure 2.** Images of 1st-6th instar larvae of FAW.

### 2.3. Image-Based Identification Method of FAW Larval Instars

First, YOLOv4 is used to locate the larvae of FAW in the image. We then use the improved segmentation model to achieve the larva region. The larval image features are extracted and the dimension reduction is performed. Finally, the improved random forest model is used to identify the larval instars. Figure 3 shows the identification procedure of FAW larval instars based on the image processing and machine learning model.

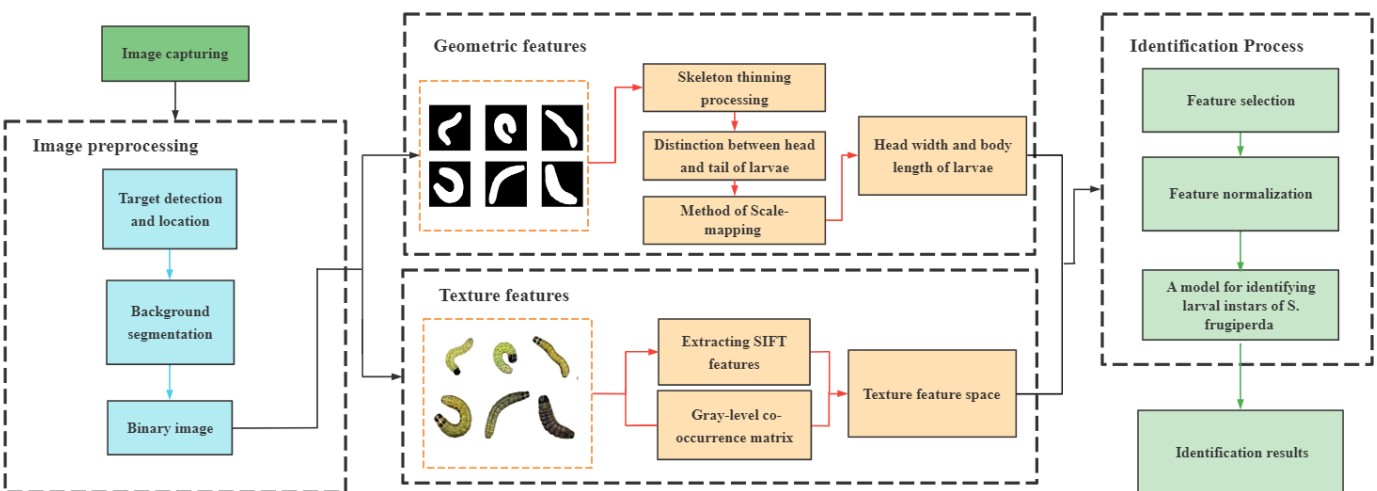

**Figure 3.** The architecture of FAW larval instar identification method.

#### 2.3.1. Location and Segmentation of Larva Region

In order to obtain a better larva segmentation performance, the target detection model YOLOV4 [24] was used to locate the FAW larvae in one image.

After locating the larva region, an improved image segmentation algorithm named MRES-UNet++ is proposed to get the larva region for the following feature extraction. The network of MRES-UNet++ is showed in Figure 4.

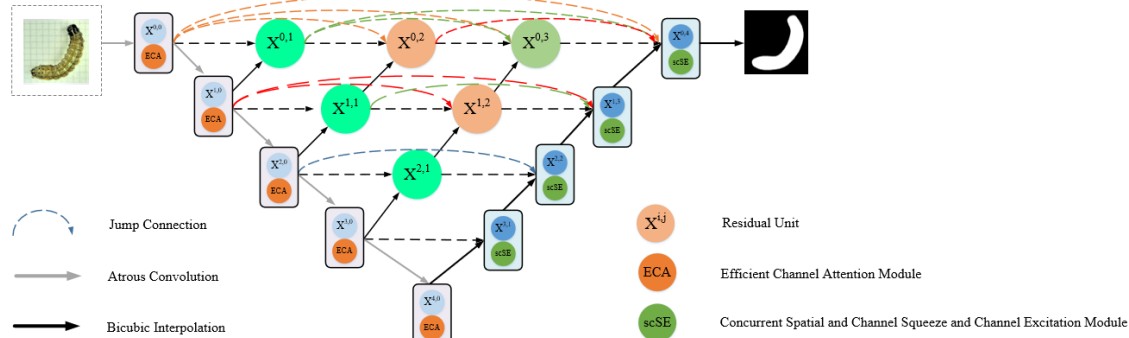

**Figure 4.** The architecture of MRES-UNet++ for segmenting the FAW larva.

To extract the larval region in images, MRES-UNet++ is improved on the basis of UNet++ [25–27], which mainly involves the preprocessing module, feature encoding (feature extraction and down sampling), and feature decoding. In the preprocessing module, in order to improve generalization ability and robustness of the semantic segmentation model, a multi-scale details enhancement algorithm [28] named Retinex is added to the original workflow, which eliminates the shadow and made details more visible. In the feature encoding stage, we replace the backbone network in the encoder with the improved ResNet34 network structure. For the purpose of reducing the information loss in the downsampling process, this paper adds three steps of dilated convolution before and after the residual structure to enlarge the receptive field. The Group Normalization (GN) layer is inserted into each convolutional layer to calculate the mean value and variance of each channel, which reduces the calculation deviation when the input parameters are normalized, so as to solve the problem of distribution changes when the parameters between layers were updated during training. The improved ResNet34 residual structure is shown in Figure 5. Furthermore, to extract more information in ROI (which is an area where target pixels may exist) in the images, this paper adds an ECA module based on effective channel attention [29] in the downsampling process. The ECA module can capture the information correlation between channels and enhance the ability to extract information in the shallow neural layer. In addition, during upsampling, the parallel module scSE [30] is added to improve the smoothness of the image contour edge when the spatial information is calibrated, which can promote the fine-grained image segmentation.

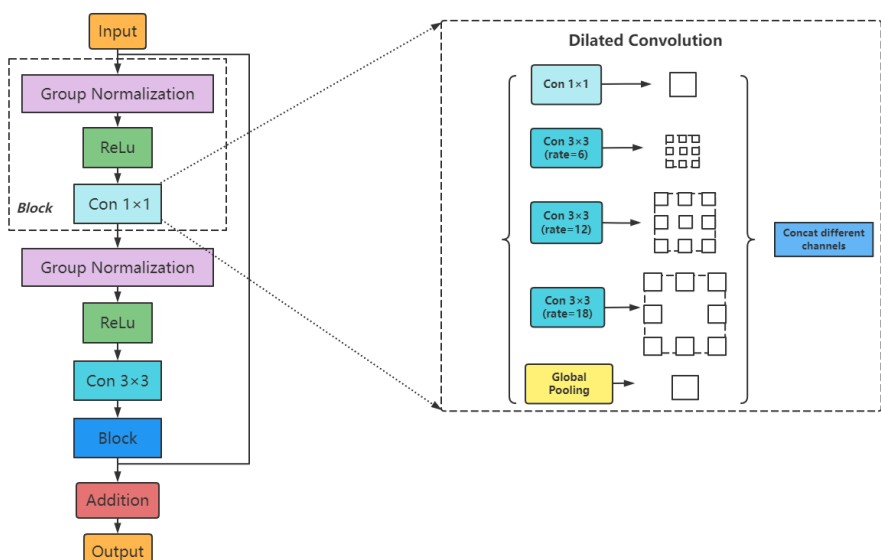

**Figure 5.** The architecture of improved ResNet34.

2.3.2. Image Feature Extraction

The FAW larvae often present the bending state after being placed on the insect container. An automatic algorithm is proposed for measuring larval body length based on the larva mask images and an optimized skeleton refinement algorithm [31,32]. First, we perform the Euclidean distance transformation to obtain the distance values of all foreground pixels and skeleton pixels. After sorting the distance values, the distance field of the image is established, and then the target corrosion operation is performed by constructing a 3 × 3 structure (kernel, one pixel block for image processing operation), and the foreground pixels are extracted from the edge. After erosion, the skeleton lines are obtained and topological relationships in image space are established. When a bifurcation point is encountered during the corrosion process, it is required to determine whether the topological relationship in the 3 × 3 grid is destroyed, that is, to remove the center point pixel and observe the connectivity of the remaining pixels. If it is not connected, the corrosion operation should continue at the bifurcation. The central axis of the larval body without bifurcation points is obtained, and the topological continuity is always maintained, which is a new mesh pruning algorithm. The comparison of the effect before and after pruning is shown in Figure 6. The actual body length of the larvae is calculated according to a square on the image with 1cm side length.

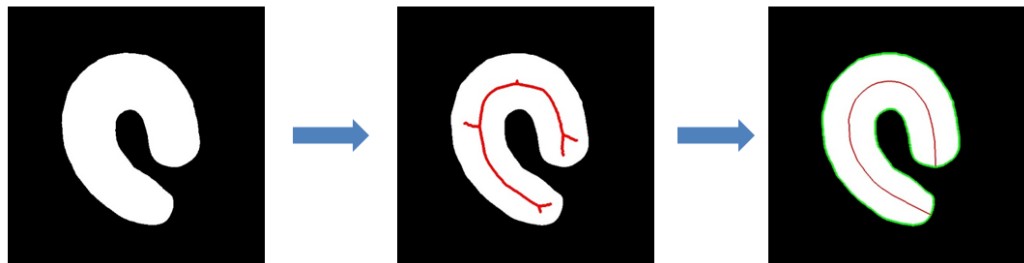

**Figure 6.** Rendering of the mesh pruning algorithm.

In order to automatically obtain the head capsule width of larvae, the MRES-UNet++ segmentation model is used to extract the binary image of the larval head. The contour is then smoothed to extract the head edge and the maximum circumscribed circle of the edge contour is drawn. The diameter is approximated as the width of the larval head capsule according to the square side length. The schematic diagram of the head capsule width calculation is shown in Figure 7.

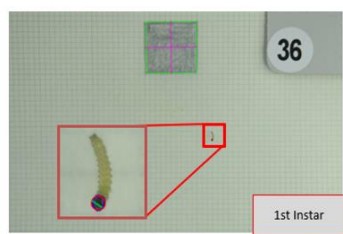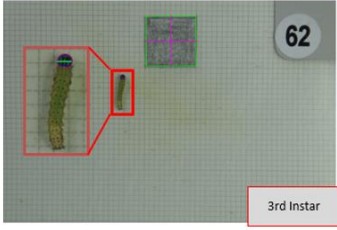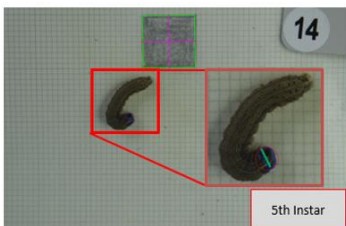

**Figure 7.** Example of calculating head capsule width.

Because the body color of FAW larvae which feed on different crops differ greatly, we extract the texture features for identifying the FAW larval instars. In order to achieve the clearer texture features, the Retinex detail enhancement algorithm [33] is used to enhance the images. In Figure 8, the texture features of the insect body are more obvious and more abundant. The Gray Level Concurrence Matrix (GLCM) [34,35] is used to extract its texture features. In order to improve the extraction speed of texture features, the gray levels are compressed from 256 to 64.

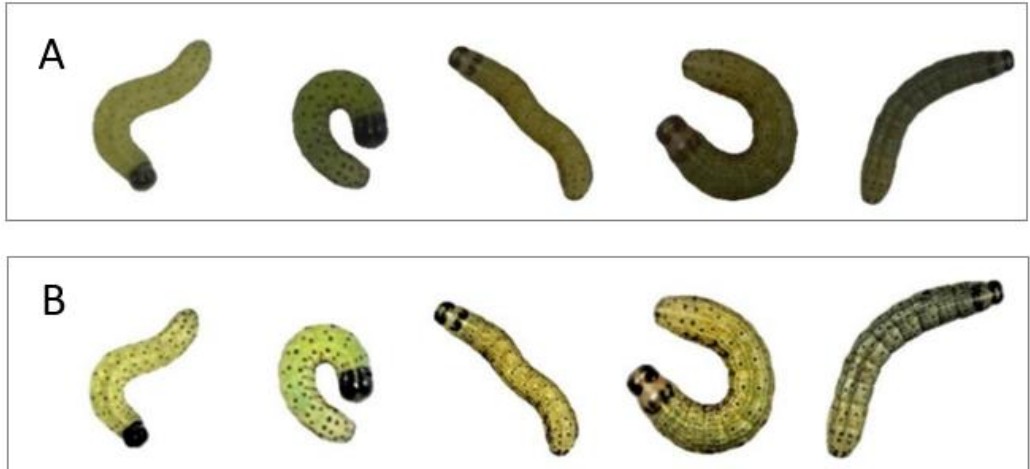

**Figure 8.** The enhancement effect of Retinex algorithm. (**A**) before enhancement; (**B**) after enhancement.

Nine grayscale matrix statistic values (mean, entropy, contrast, correlation, autocorrelation, homogeneity, and energy) are calculated, and the average value of the above four directions are taken as the texture feature for the grayscale combination to describe the grayscale relationship and spatial features between image pixels. The extraction effect of specific texture parameters is shown in Figure 9.

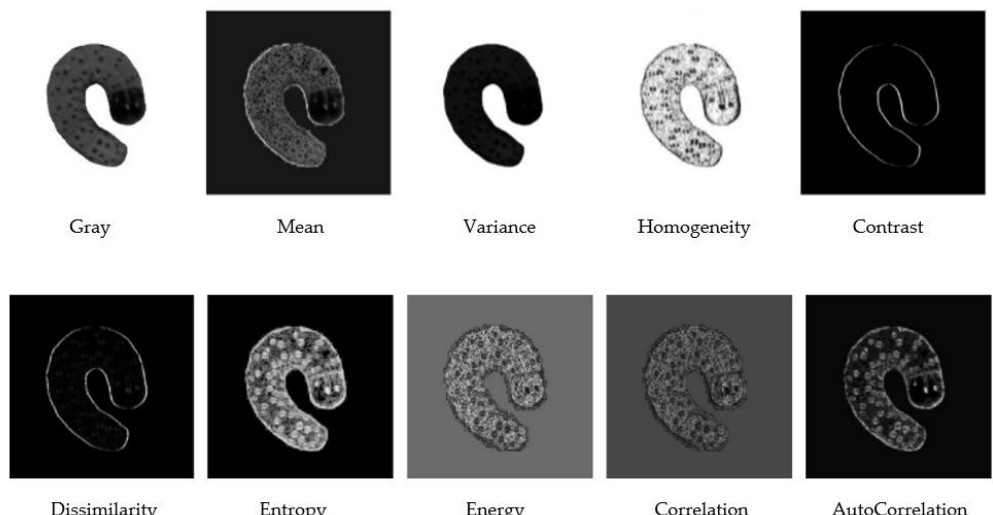

**Figure 9.** Nine texture feature maps of a FAW larva.

The SIFT features [36–38], as one of the important algorithms for image feature matching, is also commonly used in image classification with the characteristics of high robustness and fast calculation speed. Compared with single edge features or gradient features, it has stronger stability and is suitable for key point matching of multi-scale targets. It is not easily affected by illumination, target orientation, deformation differences, etc. It mainly scales the image by setting the scale factors of three scales, and then performs Gaussian blurring and subtracts the images to obtain a set of multi-scale difference images. After comparing the pixel values of the $3 \times 3$ pixel blocks of each three images to find the extreme point, it is described by the position, gradient direction and gradient size. A total of 128 dimensional features of SIFT are extracted.

### 2.3.3. Dimension Reduction of the Feature Vector

In order to improve the algorithm efficiency, the Boruta feature selection algorithm [39] is improved to reduce the dimension of the 139 dimensional feature vector. Initially,

80%, 100%, and 50% eigenvalues from original feature set *R* are extracted according to different feature types (geometry, sift) for random sorting and create a shadow feature set *S*. Then, *R* and *S* are combined to a new feature set *N* (*N* = [*R*, *S*]). Taking the set *N* as the input, the highest importance (Z-score) of all shadow features is selected as the reference value Max_shadow, and compared with the original features. Among them, the importance value which is higher than Max_shadow is marked once and retained. We set the number of iterations to conduct experiments, continuously create the new shadow features to generate new feature structures for feature importance comparison, and remove unimportant features until all remaining features are marked as passed. Finally, 44 valid features are remained.

### 2.3.4. Identification Model of FAW Larval Instars Based on an Improved Random Forest Model

In machine learning, there are many classification methods, such as logical regression, decision tree, SVM [40], random forest [41], etc. In the multi classification, the SVM and random forest algorithms are normally used. The SVM method supervises and learns the data features, finding an optimal classification hyperplane, and maximizes the interval between different classifications. By using nonlinear mapping, the linear non-separable problem in low-dimensional space is transformed into a linear separable problem in high-dimensional space to solve. The SVM method requires a complex parameter tuning process and is less interpretable. And yet the random forest algorithm does not need to adjust too many parameters, which can solve the multi classification problem well. The random forest model [41,42], as an ensemble learning of decision tree and bagging methods, has a good anti-noise ability and excellent stability. In the model training process, each decision tree is trained by randomly sampling the training set for many times, and the used eigenvalues are also randomly and not repeatedly extracted, according to a certain proportion, to ensure that each decision tree can output the stable judgment results. By injecting random disturbances, the correlation between each decision tree is decreased, and the anti-interference ability of the entire model is enhanced.

The steps of general random forest are as follows.

(1) Bootstrap resampling [43]. A certain proportion of samples are randomly selected from the training set to construct the feature subset, and the samples that are not drawn are used as the verification set to verify the correctness of the model.

(2) Decision tree generation. By Using the CART algorithm [44], partial discrete features are randomly extracted to obtain feature subsets, and the optimal features in the feature subsets are selected as decision tree nodes in each iteration so that the decision tree can continuously split and grow without pruning as much as possible.

(3) Iterative verification. Upon repeating steps 1 and 2, the number of repetitions is generally the number of decision trees. The hyperparameter space is constructed in the iterative process, and the parameters are adjusted according to the influence of the hyperparameters on the accuracy.

(4) Comprehensive voting. The voting strategy [45] of majority voting is adopted to comprehensively analyze the classification results of each decision tree, and the final training model is obtained to predict the final classification result.

In order to prevent over-fitting and enhance the identification ability and stability of the model, in step (3), the RandomizedSearchCV [46] and GridSerachCV [23] methods were proposed to optimize hyperparameters, and the number of iterations of the learner was dynamically adjusted according to the training situation. The RandomizedSearchCV algorithm [22] is used for randomly searching to narrow the parameter selection range of the hyperparameter space for a rough search, and then the GridSerachCV algorithm [23] is used for fine searching to construct the optimal hyperparameter set. The combination of the two algorithms quickly and efficiently finds the optimal parameters and expands the search scope. Furthermore, a 10-fold cross validation was conducted to optimize the hyperparameter space and the most suitable combination of hyperparameters was obtained.

In this paper, a parameter search is performed by setting the ranges of the two parameters of n_estimators ([10, 300], stride = 10; [60, 70], stride = 1), and max_depth ([1, 15], stride = 1), and finally the optimal hyperparameter combination is obtained, that is, n_estimators = 62 and max_depth = 7, making the model have the best average identification accuracy and auc, and improve average precision, average recall, and average *F1* value. The improved random forest method is shown in Figure 10.

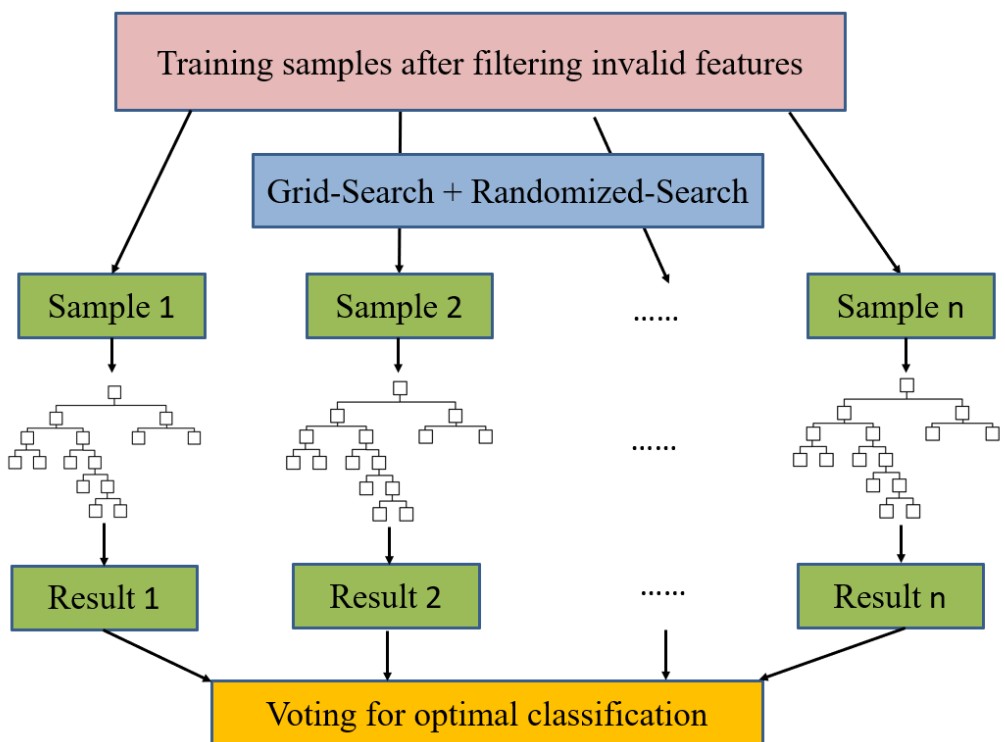

**Figure 10.** The architecture of the improved random forest model.

*2.4. Evaluation Method*

2.4.1. Evaluation Metrics for Larva Region Segmentation

The performance of the segmentation algorithm for FAW larvae was evaluated by the metrics *PA* (pixel accuracy), *MIoU* (mean intersection-over-union ratio), and *FWIoU* (Frequency Weighted Intersection over Union). The formulas are as follows.

$$PA = \frac{\sum_{i=0}^{1} p_{ii}}{\sum_{i=0}^{1} \sum_{j=0}^{1} p_{ij}} \tag{1}$$

$$MIoU = \frac{1}{1+1} \sum_{i=0}^{1} \frac{p_{ii}}{\sum_{j=0}^{1} p_{ij} + \sum_{j=0}^{1} p_{ji} - p_{ii}} \tag{2}$$

$$FWIoU = \frac{1}{\sum_{i=0}^{1} \sum_{j=0}^{1} p_{ij}} \sum_{i=0}^{1} \frac{\sum_{j=0}^{1} p_{ij} p_{ii}}{\sum_{j=0}^{1} p_{ij} + \sum_{j=0}^{1} p_{ji} - p_{ii}} \tag{3}$$

where $p_{ii}$ represents the total number of pixels when the real label is *i* and the predictive classification is also *i*. $p_{ij}$ represents the total number of pixels when the true label is *i* and the predictive classification is *j*. $p_{ji}$ represents the total number of pixels when the true label is *j* and the predictive classification is *i*.

2.4.2. Evaluation Protocol of Larval Instar Identification

In order to evaluate the effect of the improved random forest model on the larval instar identification of *S. frugiperda*, *Pre* (precision), *Rec* (recall), and *F1* (f1-score), *Acc*

(accuracy), *AUC* (area under curve) were used as the evaluation indicators of this model. The calculation formulas are shown in (4)–(8).

$$Pre = \frac{TP}{TP + FP} \tag{4}$$

$$Rec = \frac{TP}{TP + FN} \tag{5}$$

$$F1 = \frac{2PR}{P + R} \tag{6}$$

$$Acc = \frac{TP + TN}{TP + TN + FP + FN} \tag{7}$$

$$AUC = \sum_{i \in (P+N)} \frac{(TPR_i + TPR_{i-1}) * (FPR_i - FPR_{i-1})}{2} \tag{8}$$

where *Pre* represents the precision, *Rec* represents the recall or TPR (true positive rate), and *F1* denotes the F1-score, which is regarded as a harmonic mean of model precision and recall. *TP* (true positive) means the number of larvae in which the instars are correctly identified. *FP* (false positive) indicates the number of larvae in which the instars are wrongly identified. *FN* (false negative) denotes the number of larvae in which the instars are wrongly judged as other instars. *TN* (true negative) refers to the number of correctly identified larvae in other instars when calculating the recognition of certain instars in the test set. *Acc* (accuracy) indicates the proportion of correctly identified larvae in the total images. In the formula of *AUC* (area under curve), that is the area under ROC curve, which is an index to measure the performance of the model. In this formula, *P* denotes the number of positive samples *N* indicates the number of negative samples, and *i* refers to a certain iteration in the calculation process of *AUC*. *FPR* (false positive rate) denotes the proportion of wrongly identified larvae.

## 3. Results

### 3.1. Segmentation Results

In order to verify the segmentation effect of the MRES-UNet++ network model, UNet, UNet++ and DeepLabv3+ models were trained and tested on the same training and testing sets, and were compared with MRES-UNet++. The segmentation results of four models are shown in Table 1.

**Table 1.** The segmentation effect of four models.

| Models | PA (%) | MIoU (%) | FWIoU (%) |
|---|---|---|---|
| DeepLabv3+ | 95.63 | 84.62 | 92.20 |
| UNet | 95.06 | 83.29 | 91.37 |
| UNet++ | 96.10 | 87.66 | 93.65 |
| MRES-UNet++ | 98.39 | 93.82 | 96.89 |

The results show that the *PA*, *MIoU* and *FWIoU* of the UNet, UNet++ and DeepLabv3+ models are fairly close. However, the MRES-UNet++ model achieves the best effect in above four segmentation models. Compared to the UNet++, the *PA*, *MIoU* and *FWIoU* of MRES-UNet++ model increase 2.29%, 6.16% and 3.24% respectively. The segmentation results of four models and GT (ground truth) are shown in Figure 11. We can see that the segmentation effects of the MRES-UNet++ are closest to that of GT.

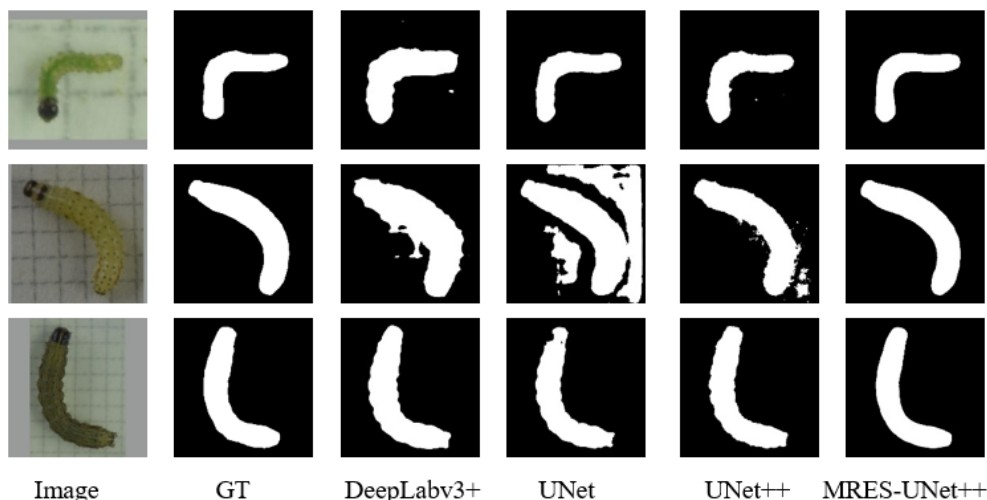

Figure 11. The segmentation results of four models for FAW larvae.

### 3.2. Result and Analysis of Larval Instar Identification

A total of 139 feature variables are obtained by using the above feature extraction methods. After feature selection, 44 effective and highly significant features were remained.

To compare with our improved random forest [42], SVM [40] and random forest models were trained and tested on the same training set and testing set. Table 2 shows the identification results of FAW larval instars by the three models.

**Table 2.** The identification results of FAW larval instars using three models.

| Instar | SVM | | | Random Forest | | | Improved Random Forest | | |
|---|---|---|---|---|---|---|---|---|---|
| | *Pre* | *Rec* | *F1* | *Pre* | *Rec* | *F1* | *Pre* | *Rec* | *F1* |
| 1 | 75.63 | 86.10 | 80.50 | 84.61 | 91.70 | 88.04 | 97.12 | 91.70 | 94.33 |
| 2 | 44.00 | 50.02 | 46.75 | 68.22 | 68.23 | 68.19 | 80.13 | 90.91 | 85.12 |
| 3 | 73.69 | 58.32 | 65.11 | 84.23 | 66.66 | 74.44 | 95.26 | 83.33 | 88.90 |
| 4 | 67.62 | 85.23 | 75.42 | 75.00 | 77.75 | 76.36 | 86.71 | 96.31 | 91.22 |
| 5 | 96.92 | 66.00 | 78.53 | 89.48 | 72.38 | 80.00 | 97.68 | 91.52 | 94.55 |
| 6 | 78.63 | 95.70 | 86.31 | 63.66 | 91.32 | 75.03 | 92.04 | 100.0 | 95.82 |
| Mean | 72.75 | 73.59 | 72.10 | 77.53 | 78.01 | 77.01 | 91.49 | 92.30 | 91.66 |

The average precision, average recall, and average *F*1 value of SVM are only 72.75%, 73.59%, and 72.10%, respectively; the differences in indicators can be more clearly observed in Figure 12A. In addition, three evaluation indexes of the 2nd instar larvae are extremely poor. And for the random forest model, the overall indicators have improved slightly, which is higher than SVM. Our improved random forest method has a precision of more than 85% in the 1st, 3rd, 4th, 5th and 6th larval instars, which is better for larval instar distinction in these three models.

In the SVM, random forest (RF) and improved random forest (RF) models, the overall accuracies are 73.71%, 85.59% and 92.22% respectively, and the *AUC* values reach 84.25%, 82.34% and 95.31% respectively, which can be observed in Figure 12B.

The Roc curve in Figure 13 describes the identification ability of each model. If each curve is closer to the left, the larger the area enclosed by the horizontal and vertical coordinates is, and the stronger the identification ability of the model is.

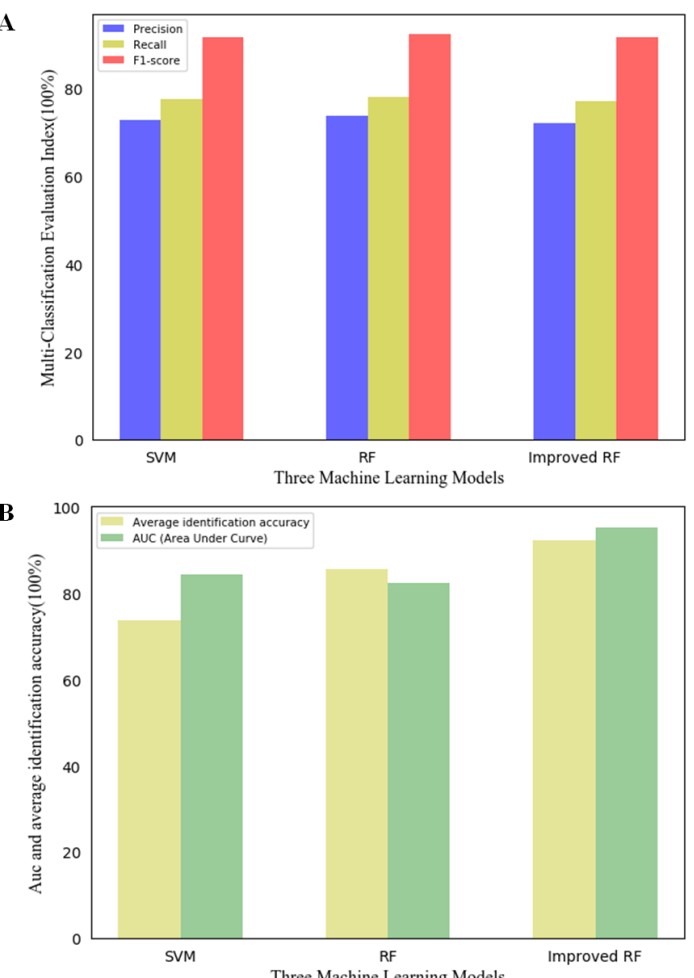

**Figure 12.** The difference in *F1*, *Pre* and *Rec* (**A**) and comparison of *AUC* and Average identification accuracy (**B**).

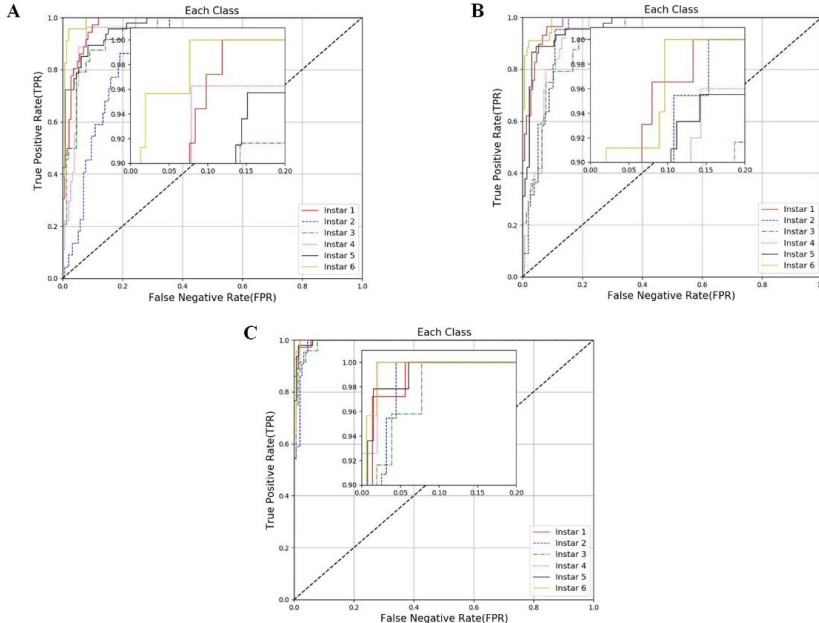

**Figure 13.** The ROC curves of SVM (**A**), random forest (**B**) and improved random forest (**C**) models.

In addition, after hyperparameter tuning and feature dimensionality reduction, the size and recognition speed of the models also change accordingly. The size of the improved random forest model is optimized to 1.16 MB, and the recognition speed reaches 36.14 ms/per image. Compared with the original random forest model, the size is reduced by 32.56% and the speed is increased by 63.45%. This is clearly presented in Figure 14.

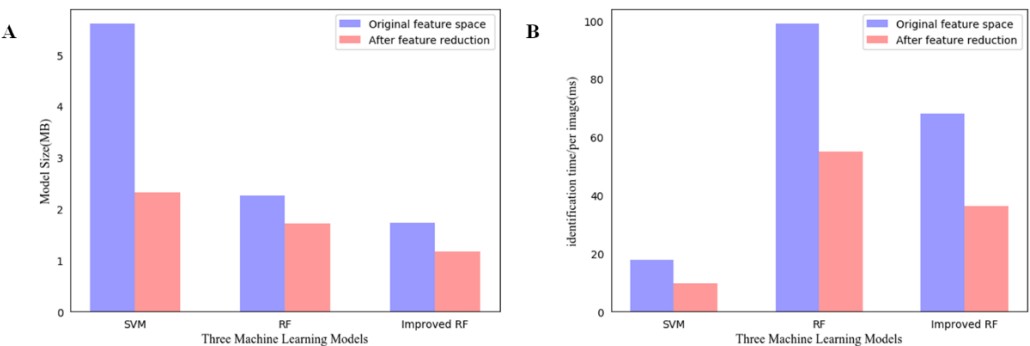

**Figure 14.** Size comparison of different machine learning models (**A**), random forest (**B**) Comparison of the identification speed of different models for each image.

## 4. Discussion

In this study, we design a low-cost image capturing device and propose an automatic larval instar identification method. The semantic segmentation network combined with improved graphics algorithm is used to realize the automatic detailed calculation of the larval head capsule width and body length, and achieving an *MIoU* of 93.82% and controlled the calculation error within 5%. Meanwhile, the Boruta dimension reduction and double search (RandomizedSearchCV [46] and GridSerachCV [23] algorithm) are used to prevent the overfitting phenomenon and high calculation costs during model training. Through these optimizations, we control the size of the model to 1.16 MB and improve the identification speed to 36.14 ms per image.

The research on larval development growth was done by stereo microscope [47,48] or hyperspectral imaging technology [49,50]. Although the insect images collected with these devices often have high definition, the hardware cost was high and professional operators were required. However, manual operation often produced some misoperations, and the efficiency was low. Our portable image capturing device uses a mobile phone with a macro lens, which does not need the manual estimation or measurement of larval instars. We randomly place one larva from the 1st–6th instar to take photos, skipping the step of adjusting the worm posture in the traditional method. The larval body does not need to be moved to obtain the ideal posture, which reduces the time consumed by traditional measurement and moving of the larvae. Compared with the previous methods, it has great portability and significantly reduces the hardware cost.

In addition, before the determination of the larval instar, some segmentation algorithms were often used to segment the larval body to achieve feature extraction and calculation. There were some examples, such as the semi-automatic interactive segmentation algorithm based on Graphcut [51], the region growing segmentation algorithm [52] based on topology simulation, and the segmentation algorithm [25–27] based on deep learning. Because of the need of automatic segmentation, we optimized the UNet++ algorithm in semantic segmentation to improve the information extraction ability of feature network and edge smoothness. In the network structure, the addition of attention mechanism often has a good effect on the extraction of feature details. In this paper, we have adopted some optimization strategies to improve the integrity of larval segmentation. In the down sampling process, the addition of eca attention module [29] uses 1D convolution instead of an FC layer to effectively capture the channel information between different feature layers. The introduction of atrous convolution increases the receptive field and expands the range of information extraction. The bicubic interpolation method and the application of scsE

attention [30] in the process of up sampling greatly reduces the information loss in pixel reconstruction, and have a certain smoothing effect on the segmentation edge.

In machine learning, the multi-classification of targets often has some problems, such as high feature complexity, difficulty in finding optimal parameters, high model complexity, and unstable performance. In view of the above situation, the feature dimension reduction optimization strategy and parameter optimization search method adopted in this paper can effectively solve the redundancy of input features and the search for optimal hyperparameters. Meanwhile, in order to verify the stability of the model, this paper demonstrates the performance improvement after model optimization by comparing the ROC curves of SVM, Random Forest and the improved Random Forest, and by comparing indicators such as *Auc*, *Pre*, *Rec*, *F*1.

In addition, this paper combines the automatic segmentation method of deep learning with the identification method of machine learning, which effectively merges the advantages of both. It is difficult to measure the size of living larvae manually, especially for the low instar larva. The segmentation model based on the deep learning method often has stable fine segmentation ability, which is convenient for automatic image segmentation and has great advantages in practical applications. However, the random forest [22,41,42] method in machine learning can adapt to different data sets, and the algorithm runs faster, and require only a few parameter optimizations. The combination of the two can often produce the best identification effect. With the expansion of image data and the improvement of hardware equipment, these models can also be used in more complex outdoor scenes in the future. However, although our algorithm has achieved good identification results, there are still some disadvantages. Although the improved random forest model improves the average identification accuracy and performance, it still takes more time than SVM. However, there is no doubt that this problem can be solved by collecting more images, optimizing the training network and improving the search strategy for hyperparameters.

## 5. Conclusions

To achieve automatic and accurate larval instars identification of *S. frugiperda* in fields, this paper designs an image acquisition device and a larval instar identification method. The device is composed of a smart mobile phone with a macro lens and an insect container with grid paper. The technician can easily capture images of larvae with this device in fields. For obtaining the morphological and textural features, we firstly locate the larval region by the YOLOv4 detection, MRES-UNet++ segmentation method and feature extraction methods. The improved random forest model is used to identify the larval instars of FAWs and achieves 92.22% and 95.31% in overall accuracy and auc respectively. Our method provides an idea for larval instar identification during pest field investigations. The device is portable and the identification method of larval instars is accurate and automatic. The non-professional person can easily use it. Although this method has carried out a detailed identification of the larval instar to a certain extent, there are some errors in the larval segmentation of larvae and the digital calculation of features, and there are still some instar misjudgments among low instar larvae. Thus, it can be considered to collect more images to improve the IoU of the segmentation model and the accuracy of the identification model, and optimize the hardware device, such as by adding ultra-wide-angle lenses to increase the clarity of the captured images and the range of light exposure.

**Author Contributions:** Methodology, J.X.; Validation, B.Y. and Q.Y.; Investigation, Z.F. and B.Y.; Resources, B.Y.; Writing—Original Draft Preparation, J.X.; Writing—Review and Editing, J.X., Q.Y. and B.Y.; Funding Acquisition, Q.Y.; Data Curation, J.T. and S.L.; Project Administration, Z.D. and J.L. All authors have read and agreed to the published version of the manuscript.

**Funding:** The research was supported by Zhejiang Provincial Natural Science Foundation of China (No.LY20C140008) and Nanjing Customs Research Project (2021KJ43).

**Institutional Review Board Statement:** Not applicable.

**Data Availability Statement:** The data presented in this study are available in the article.

**Conflicts of Interest:** The authors declare that they have no conflict of interest.

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
