# Peer review of "Improved Random Forest for the Automatic Identification of Spodoptera frugiperda Larval Instar Stages"

_agriculture, doi:10.3390/agriculture12111919_

Round 1

Reviewer 1 Report

The authors designed a portable image acquisition device using a mobile phone with a macro lens to collect 1st-6th instar larva images. The YOLOv4 detection and improved MRES-UNet++ segmentation methods were used to locate the larvae and segment the background. The research is critical and relevant however, the presentation and narration must be improved to be easily understood by the general scientific audience. The following is what was observed:

Every study is research; hence, I think the title must be altered: Suggested title is Improved random forest for the automatic identification of Spodoptera frugiperda larval instars stages.

The introduction must fully motivate the need for such a study as well as show the gap and innovation of the current study.

The methodology is not elaborately described to be repeatable; it is too abstract to be easily understood. For instance:

What was improved on the random forest, why was it improved, and how did it differ in performance from the normal random forest? Why was the random forest selected and not any other machine-learning method? This must be clear in the introduction as well as methodology sections to allow a smooth flow of ideas.

line 136 ‘Furthermore, to extract 136 more interest information in the images, this paper added an ECA module based on effective ….’ What interest information do the authors mean?

Line 151 why was the 3 x 3 structure used?

Line 221 Which hyperparameters were used and what were the thresholds or what determines that these are the optimum values?

There is no justification with literature support of why steps 1-4 presented on lines 224-239 are important. There is a need to justify each choice of method and compare it to available literature to demonstrate the addition to science as well as to show the novelty. However, this is missing in the descriptions in the entire document.

The normal random forest and SVM are only introduced in the results but nothing in the methodology.

It is not clear how the lens of the mobile device influences the results and how this was factored into the analysis. Was there a special lens used? If so, how was it calibrated, for the purpose of proof of concept as well as for general use? How will the proposed methodology be affected by the type and quality of mobile devices since the design is for different technicians in the field?

It is not clear if the algorithm was designed in the form of a mobile application or which programming language was used.

Discussion 315-352 seems like just a narration of the methodology without really relating the results to the purpose of the developed methodology. It was expected that the authors could have dived in depth with the merits and demerits of the methodology to allow better understanding and relationship with already available similar concepts.

If this methodology is to be acceptable and improved as the authors claim, authors must demonstrate without a doubt that the method is foolproof but at the present time, there is no convincing account except a narration.

Several grammatical, and spelling errors were observed, and an English editor is recommended. 

Author Response

Dear Editors and Reviewers:

Thank you for your letter and the Editor comments on our manuscript entitled“Research on automatic identification method of Spodoptera frugiperda larval instars based on improved random forest model” (agriculture-1983927). Those comments are very helpful for revising and improving our paper, as well as the important guiding significance to our research. We have studied the Editor comments carefully and made corrections which we hope meet with approval. Revised portions are marked in red on the paper. The main corrections are in the manuscript and the responses to the comments are as follows:

Point 1. Every study is research; hence, I think the title must be altered: Suggested title is Improved random forest for the automatic identification of Spodoptera frugiperda larval instars stages.

Response 1: In lines 2. According to your suggestions, we have made corresponding modifications to the title.

Point 2. The introduction must fully motivate the need for such a study as well as show the gap and innovation of the current study.

Response 2: In lines 46. In the introduction, we have added a description of research needs. Line85.We discussed the innovation of research in detail.

Point 3. What was improved on the random forest, why was it improved, and how did it differ in performance from the normal random forest? Why was the random forest selected and not any other machine-learning method? This must be clear in the introduction as well as methodology sections to allow a smooth flow of ideas.

Response 3: In lines 85. Random Forest method has the advantages of low parameter adjustment complexity and low computational cost for multi classification applications, which is suitable for current research. The improved method mainly reduces the number of input features through Boruta dimension reduction for improved sampling, reduces the complexity of the model, and accelerates the training speed of the model through RandomizedSearchCV and GridSerachCV strategies. Based on your suggestions, we have added corresponding instructions to this part in the introduction.

Point 4. line 136 ‘Furthermore, to extract 136 more interest information in the images, this paper added an ECA module based on effective ….’ What interest information do the authors mean?

Response 4: In lines 159.interest information, that is, information about the region of interest (ROI). It is an area where target pixels may exist. ROI is a technical term in the field of deep learning.

Point 5. Line 151 why was the 3 x 3 structure used?

Response 5: In lines 173. 3 x 3 structure is a pixel block for image erosion. The structure of 5 * 5 will cause excessive corrosion of the target's central axis and premature fracture at the bifurcation, while 1 * 1 will cause insufficient corrosion, there is still bifurcation, which affects the length calculation. Only the size of 3 * 3 structure can perfectly obtain the standard larval length.

Point 6. Line 221 Which hyperparameters were used and what were the thresholds or what determines that these are the optimum values?

Response 6: In lines 273.We used n_estimators and max_ depth ([10,300], stride=10; [1,15], stride=1). The hyperparameters are optimal when the auc and average identification accuracy are the highest.

Point 7. There is no justification with literature support of why steps 1-4 presented on lines 224-239 are important. There is a need to justify each choice of method and compare it to available literature to demonstrate the addition to science as well as to show the novelty. However, this is missing in the descriptions in the entire document.

Response 7: In lines 249.We have rearranged this part of logic for readers to understand, and added some method descriptions and some references to prove the necessity of method selection.

Point 8. The normal random forest and SVM are only introduced in the results but nothing in the methodology.

Response 8: In lines 234.We have added the description of this part in the method, but this paper more focuses on the method of the improved part.

Point 9. It is not clear how the lens of the mobile device influences the results and how this was factored into the analysis. Was there a special lens used? If so, how was it calibrated, for the purpose of proof of concept as well as for general use? How will the proposed methodology be affected by the type and quality of mobile devices since the design is for different technicians in the field?

Response 9: In lines 109.On the hardware side, the macro lens is spliced with 4.3x magnification respectively. The mobile phone can achieve automatic focus. This method is applicable to mainstream mobile phones on the market, which is not affected by the device type.

Point 10. It is not clear if the algorithm was designed in the form of a mobile application or which programming language was used.

Response 10: We use python language to program the algorithm, and demonstrate strength of the algorithm through experiments.

Point 11. Discussion 315-352 seems like just a narration of the methodology without really relating the results to the purpose of the developed methodology. It was expected that the authors could have dived in depth with the merits and demerits of the methodology to allow better understanding and relationship with already available similar concepts.

Response 11: In lines 352.We have reorganized and combed some of the contents of the discussion, making the results more closely related to the discussion.

Point 12. If this methodology is to be acceptable and improved as the authors claim, authors must demonstrate without a doubt that the method is foolproof but at the present time, there is no convincing account except a narration.

Response 12: In lines 354.We modified the logic of the discussion part, and demonstrated the reliability of the method in detail through supplementary explanations. At the same time, we added corresponding references to increase support.

Point 13. Several grammatical, and spelling errors were observed, and an English editor is recommended.

Response 13: We re-checked and corrected English grammatical and spelling errors in more detail, which were marked as red.

Reviewer 2 Report

The article includes a new method for the intelligent, precise and automatic identification of the larval instars of Spodoptera frugiperda (fall armyworm, FAW). This topic is original and relevant in its field. The methodology is correctly exposed, and the results response the main question posed. The number of references is appropriated for a research article, but some of them are very old. Try to avoid including references older than 20 years, except in some justified cases.

Therefore, some comments must be addressed to improve the article:

- The text must be completely adapted to the journal template.

- Section Introduction. Include some references on the limits of the practical use of the image processing technology and automatic segmentation methods

- Section Materials and Methods. The image capturing device must be described.

- The labels of Figure 12 are very small for easy reading.

- Section Discussion. A proper discussion of results obtained must be included.

- Section Conclusion. A reflection must be made on the practical use and the limits of the applications on the proposed method.

- Reference 6. Mention the language of the article in the reference.

Author Response

Dear Editors and Reviewers:

Thank you for your letter and the Editor comments on our manuscript entitled“Research on automatic identification method of Spodoptera frugiperda larval instars based on improved random forest model” (agriculture-1983927). Those comments are very helpful for revising and improving our paper, as well as the important guiding significance to our research. We have studied the Editor comments carefully and made corrections which we hope meet with approval. Revised portions are marked in red on the paper. The main corrections are in the manuscript and the responses to the comments are as follows:

Point 1. The number of references is appropriated for a research article, but some of them are very old. Try to avoid including references older than 20 years, except in some justified cases.

Response 1: In lines 456. First of all, we have replaced some old references based on your suggestions

Point 2. The text must be completely adapted to the journal template.

Response 2: In lines 283.We adjusted some formats to meet the requirements of the journal, such as some formatting in formulas and forms.   

Point 3. Section Introduction. Include some references on the limits of the practical use of the image processing technology and automatic segmentation methods

Response 3: In lines 80. For these suggestions, we discuss some limitations of current intelligent instar identification methods. And a brief overview of the current improved random forest method and semantic segmentation method, leading to follow-up content.

Point 4. Section Materials and Methods. The image capturing device must be described.

Response 4: In lines 109.For your suggestions, we have added the description of hardware equipment.

Point 5. The labels of Figure 12 are very small for easy reading.

Response 5: In lines 362.In view of this, we have resized some label styles to obtain higher clarity.

Point 6. Section Discussion. A proper discussion of results obtained must be included.

Response 6: In lines 366.We rearranged the logic of the discussion part to make it more consistent with the research results and research purposes.

Point 7. Section Conclusion. A reflection must be made on the practical use and the limits of the applications on the proposed method.

Response 7: In lines 437.We have added limitations in the practical application process to the conclusion.

Point 8. Reference 6. Mention the language of the article in the reference

Response 8: In lines 468. We have changed the language of the references to English to make it easier for readers to understand.

Round 2

Reviewer 1 Report

All comments were addressed. Check the spelling of 'GridsearchCV' if it is correct.